# Phenological Responses to Snow Seasonality in the Qilian Mountains Is a Function of Both Elevation and Vegetation Types

**Yantao Liu** [1], **Wei Zhou** [1,2,3,*], **Si Gao** [1], **Xuanlong Ma** [4] and **Kai Yan** [1]

1   School of Land Science and Technology, China University of Geosciences Beijing, Beijing 100083, China; ytliu@cugb.edu.cn (Y.L.); gaosi_2021@email.cugb.edu.cn (S.G.); kaiyan@cugb.edu.cn (K.Y.)
2   Key Laboratory of Land Consolidation and Rehabilitation, Ministry of Natural Resources, Beijing 100035, China
3   Technology Innovation Center for Ecological Restoration in Mining Areas, Ministry of Natural Resources, Beijing 100083, China
4   College of Earth and Environmental Sciences, Lanzhou University, Lanzhou 730000, China; xlma@lzu.edu.cn
*   Correspondence: zhouw@cugb.edu.cn

**Abstract:** In high-elevation mountains, seasonal snow cover affects land surface phenology and the functioning of the ecosystem. However, studies regarding the long-term effects of snow cover on phenological changes for high mountains are still limited. Our study is based on MODIS data from 2003 to 2021. First, the NDPI was calculated, time series were reconstructed, and an SG filter was used. Land surface phenology metrics were estimated based on the dynamic thresholding method. Then, snow seasonality metrics were also estimated based on snow seasonality extraction rules. Finally, correlation and significance between snow seasonality and land surface phenology metrics were tested. Changes were analyzed across elevation and vegetation types. Results showed that (1) the asymmetry in the significant correlation between the snow seasonality and land surface phenology metrics suggests that a more snow-prone non-growing season (earlier first snow, later snowmelt, longer snow season and more snow cover days) benefits a more flourishing vegetation growing season in the following year (earlier start and later end of growing season, longer growing season). (2) Vegetation phenology metrics above 3500 m is sensitive to the length of the snow season and the number of snow cover days. The effect of first snow day on vegetation phenology shifts around 3300 m. The later snowmelt favors earlier and longer vegetation growing season regardless of the elevation. (3) The sensitivity of land surface phenology metrics to snow seasonality varied among vegetation types. Grass and shrub are sensitive to last snow day, alpine vegetation to snow season length, desert to number of snow cover days, and forest to first snow day. In this study, we used a more reliable NDPI at high elevations and confirmed the past conclusions about the impact of snow seasonality metrics. We also described in detail the curves of snow seasonal metrics effects with elevation change. This study reveals the relationship between land surface phenology and snow seasonality in the Qilian Mountains and has important implications for quantifying the impact of climate change on ecosystems.

**Keywords:** land surface phenology; NDPI; Qilian Mountains; snow cover; high elevation

## 1. Introduction

Evidence suggests that global temperatures have continued to rise over the last two decades and will continue to warm over the next three decades [1], which affects many ecosystems [2–4]. Alpine ecosystems are considered to be particularly sensitive to climate change because of harsh natural environments [5–8]. Therefore, accurate assessment of the impacts of climate change in alpine ecosystems is essential.

Land surface phenology (LSP) is defined as the seasonal change pattern of surface vegetation obtained from remote sensing observations, which is usually used to describe

the start, end and length of the vegetation growing season [9–11]. Unlike traditional ground-based observations that can record dates of budburst or flushing, LSP is used to describe the full process of regional greening. This may not correspond to a specific vegetation event but can provide a rapid understanding of the key stages of the overall greening of a region [9]. LSP is one of the most sensitive and easily observable nature features when analyzing the response of vegetation to climate change [12], and exploring its changes provides an important avenue for studying severe climate anomalies. Climate change can interfere with vegetation germination. In addition, changes in LSP may have a significant impact on carbon and water cycles [13–15]. An integrated analysis of the impact of climate change on land surface phenology is important for understanding the impact of future climate change.

Driving factors for trends in land surface phenology have frequently been attributed to changes in temperature and precipitation [16–18]. However, as one of the typical features in areas with stable snowpack, changes in snow seasonality can also cause changes in land surface phenology [19,20]. Specifically, Snow accumulates or melts on soil and vegetation, which can directly alter the hydrothermal conditions under which vegetation grows and develops (Figure 1). Snow directly affects near-surface temperatures in several ways. Snow cover in winter insulates the soil from cold air and maintains soil temperature [20]. Soil temperature is higher than air temperature in early spring under snow cover [21]. Due to the insulating nature of snow, temperature no longer has a direct effect on vegetation [22]. The timing of snowmelt is sometimes a more important factor in the growing season than air temperature [23]. These regulatory effects of snow accumulation and snowmelt on surface temperature have important implications for land surface phenology and soil moisture content. Snow is the main source of freshwater for alpine vegetation, as snowmelt provides the necessary moisture for vegetation to sprout in the form of soil water [24–27]. Snow cover protects vegetation and soil from harsh natural hazards such as wind erosion, freezing damage, and intense solar radiation, which often occur in high-elevation mountains and can seriously hinder vegetation growth [28–31]. It has also been demonstrated that winter snow can indirectly affect the carbon sequestration capacity of vegetation by altering community structure and activity of soil microorganisms [32].

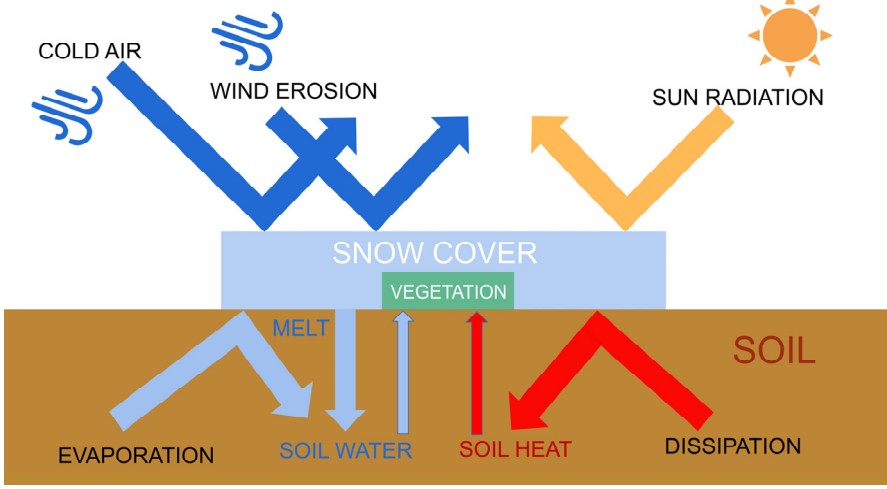

**Figure 1.** Schematic diagram of snow cover effects.

Although there are many studies focusing on the snow cover effect on land surface phenology, they have inconsistent results [33–38]. In some warm and dry regions, winter snow cover has been shown to favor early spring vegetation germination and prolong the growing season, while the opposite is true in cold, humid regions [35,36], clearly demonstrating that the effect of snow cover varies under different hydrothermal conditions. However, in Inner Mongolia, China [38], and the French Alps [39], two regions with different natural conditions, snow has a similar negative effect on land surface phenology:

late snowmelt delays vegetation emergence. Grass is the predominant vegetation type in these two regions, and the mechanism for the effect of snow on the same species should be consistent. Even so, the impact of snow on a single vegetation type in the same area changes with terrain [33,34]. In short, the influence of snow seasonality on land surface phenology is determined by the coupling of multiple factors such as water and heat conditions, vegetation type and terrain. There remains a lot of views on specific conclusions, especially the effect of snow as a function of elevation gradients and vegetation types, and additional study is required. Additionally, the spring phenology extracted using NDVI is often affected by preseason snow, which may lead to inaccurate conclusions [40,41]. Unraveling the effects of snow on land surface phenology can help identify the mechanisms of change in land surface phenology.

The Qilian Mountains area (QLMA) is an essential part of the Qinghai–Tibet Plateau and is considered as an important ecological barrier in western China. The QLMA has large elevation differences, a variety of vegetation types and significant climate change. In addition, the relatively small area of QLMA could mitigate the impact of spatial differences on the results. It is an ideal laboratory for studying the vegetation response to climate change. Studies on vegetation phenology for the Qilian Mountains region are still limited. We quantified the response of land surface phenology metrics to different snow seasonality metrics from 2002 to 2021. Annual snow seasonality metrics include the first snow day (FSD), last snow day (LSD), snow season length (SSL) and total number of days with snow cover (SCD). Land surface phenology metrics include the start of the growing season (SOS), end of the growing season (EOS) and length of the growing season (LOS) estimated by the normalized difference phenology index (NDPI), which was proven to be an accurate vegetation index for estimating land surface phenology at high elevation [42]. More comprehensive metrics and more reliable vegetation indices enhance the richness and accuracy of our conclusions. In addition, we chose two representative influencing factors, elevation and vegetation type, which may be helpful to explain the complex effects of snow seasonality.

To better understand the relationship between snow and land surface phenology, the following three research questions are proposed:

1.  What are the distribution characteristics of snow seasonality and land surface phenology in the Qilian Mountains area?
2.  What is the impact of snow on land surface phenology in the study area?
3.  How does the phenological response change with elevation and by vegetation types?

## 2. Materials and Methods

### 2.1. Study Area

The QLMA lies at the intersection of the Tibetan, Mongolian and Loess plateaus (35.84°–39.97°N, 93.61°–103.90°E), with Qinghai Province in the south and Gansu Province of China in the north (Figure 2). The average elevation of the QLMA is over 3000 m, higher in the center and lower in the surroundings. This area belongs to the mid-latitude high elevation region. Most of the QLMA is in the temperate semiarid zone of the highlands [43], where solar radiation is strong. The average annual precipitation is 300–500 mm, more in the east than in the west. These make QLMA a sensitive area for climate change.

As an important ecological barrier in northwestern China, the QLMA is home to headwaters of rivers fed by snow and glacial meltwater. Grassland vegetation, desert vegetation, shrub and alpine vegetation cover more than 90% of the QLMA. Additional vegetation types include coniferous forest, broadleaved forest and swamp vegetation [44,45]. Because of its rich gene pool of alpine species, the QLMA is considered a key area and a priority area for biodiversity conservation in China [46]. A pilot Qilian Mountain National Park in the QLMA was established in 2017 to protect forests, grasslands, wetlands, surface water resources and glaciers [47]. Collectively, the QLMA is an ideal laboratory for studying climate change and ecosystems in cold and arid regions.

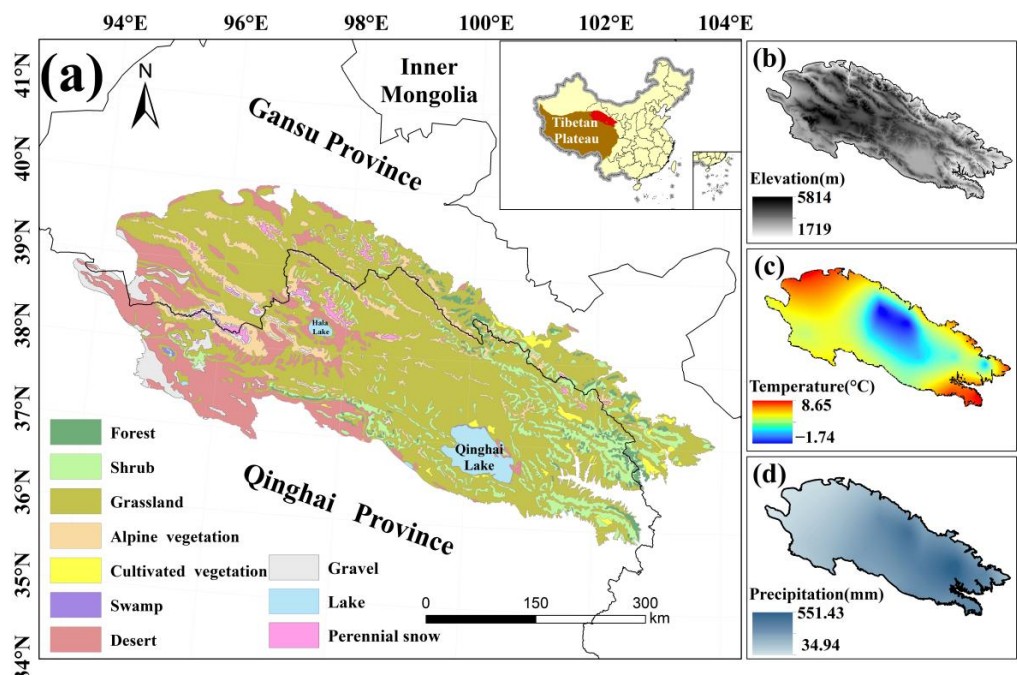

**Figure 2.** Map of the (**a**) land cover types, (**b**) elevation, (**c**) annual mean temperature, and (**d**) annual mean precipitation of the study area.

### 2.2. Data Sources and Pre-Processing

The normalized difference snow index (NDSI) was used to create snow pixels with a resolution of 500 m provided by MO(Y)D10A1 V6. The Terra(O) and Aqua(Y) satellites provide a separate daily NDSI from 1 September 2002 to 1 June 2020 [48]. 'NDSI_Snow_Cover_Class' and 'NDSI_Snow_Cover_Algorithm_Flags_QA' are the bands we used to eliminate invalid pixels and 'NDSI_Snow_Cover' is the band to extract NDSI. Data were from NASA's NSIDC DAAC at CIRES, with a large percentage of cloud pixels [49–51]. Some preprocessing was required to eliminate cloud obscuration [52,53]. First, pixels with a non-zero value of the 'NDSI_Snow_Cover_Class' and 'NDSI_Snow_Cover_Algorithm_Flags_QA' band were masked out, as these pixels represent invalid values such as missing data and clouds. Second, daily MOD10A1 and MYD10A1 data were combined using daily maximum values to reduce the number of cloud pixels. Finally, the max of the cloud-free pixels within a three-day time window surrounding the cloud pixels was taken as a replacement value. Experiments proved that these operations can effectively reduce the proportion of cloud pixels and improve the accuracy of snow products [51,54]. These preprocesses were completed using the Google Earth Engine.

The MOD09A1 V6 products provide the surface spectral reflectance of Terra MODIS bands 1–7 at 500 m resolution [55] containing seven bands that have been corrected for atmospheric conditions such as gasses and aerosols. It was used to calculate NDPI and thus estimate land surface phenology from 2003 to 2021. Data were provided by NASA LP DAAC at the USGS EROS Center.

The QLMA vegetation distribution dataset (vegetation pattern data (1:1,000,000) in the Qilian Mountains) was obtained from the National Cryosphere Desert Data Center (NCDC, http://www.ncdc.ac.cn/, accessed on 6 September 2021) and the digital elevation model (DEM) from 30 m ASTER GDEM was downloaded from the Geospatial Data Cloud (http://www.gscloud.cn/, accessed on 13 January 2022). We used these data to distinguish the six types of vegetation and every 100 m in elevation of QLMA. Figure 3 shows the technical workflows.

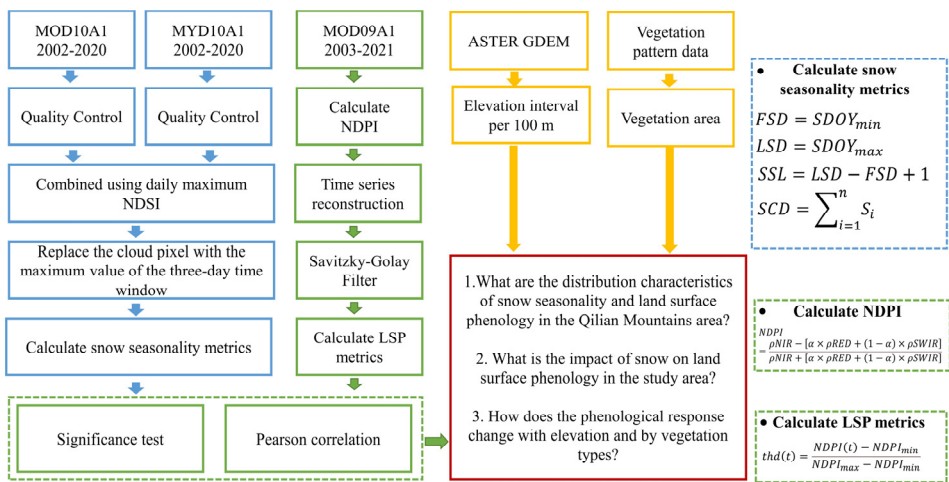

**Figure 3.** Overview of technical workflows.

### 2.3. Calculation of Snow Cover Seasonality

Value equal to 40 in the preprocessed 'NDSI_Snow_Cover' layer was chosen as a threshold for identifying snow pixels [56]. Pixels recognized as snow pixels were reclassified as 1, otherwise 0. Previous studies have shown that the snowpack on the Tibetan Plateau increases sharply in September and decreases sharply in the beginning of May [57]. Therefore, 1st September to 1st June of the following year is considered to be the potential snow season to exclude outliers that appear in the summer. A new time series was constructed: 1st September was set to be the first day of year, and 31st May of the following year was set to be the last. The metrics of the snow season were calculated based on methods in [58], where *FSD* and *LSD* are measured in day of year (DOY) and *SSL* and *SCD* are measured in days. The formulas are as follows:

$$FSD = SDOY_{min}, \tag{1}$$

$$LSD = SDOY_{max}, \tag{2}$$

$$SSL = LSD - FSD + 1, \tag{3}$$

$$SCD = \sum_{i=1}^{n} S_i, \tag{4}$$

where *SDOY* represents the distance between the date when a pixel is identified as a snow pixel and previous 1st September, *n* denotes the total number of days in the potential snow season, and $S_i$ denotes the state of snow cover for any given day within *SDOY*; $S_i$ equals 1 if there is snow and 0 if not.

### 2.4. Calculation of Land Surface Phenology

Shortwave infrared reflectance was combined with near-infrared and red reflectance to calculate the *NDPI*. It is an index for extracting accurate surface phenology to achieve high contrast between vegetation and background [59]. It was proven to be an accurate vegetation index for estimating land surface phenology in QLMA [42]. The formula is as follows:

$$NDPI = \frac{\rho NIR - [\alpha \times \rho RED + (1 - \alpha) \times \rho SWIR]}{\rho NIR + [\alpha \times \rho RED + (1 - \alpha) \times \rho SWIR]}, \tag{5}$$

where $\rho NIR$ represents the near-infrared band, $\rho RED$ represents the red band and $\rho SWIR$ represents the shortwave infrared band. In MOD09A1, $\rho RED$ corresponds to band 1, $\rho NIR$ to band 2 and $\rho SWIR$ to band 6. $\alpha$ is a constant value for a given sensor and is taken as 0.74 for MODIS products [59].

In this study, we first masked the pixels of cloud shadow, snow and cloud, then calculated *NDPI* values for the filtered images. Then we reconstructed the time series

to unify the number of images into 72 in each year by calculating the average value. Finally, we smoothed the curves using the Savitzky–Golay (SG) filter. The window size was set to 5 and the number of polynomials was set to 3. Several methods have been developed to detect land surface phenology based on vegetation indices [60–63]. A simple and effective dynamic threshold method was used to extract land surface phenology from the reconstructed *NDPI* time series (Figure 4) [64]. Pixels with an intra-year *NDPI* change of less than 0.1 were excluded, and these were assumed to be areas without significant seasonality, as the *NDPI* of vegetation with distinct phenological stages generally increases from very small values (close to 0) to above 0.4 in study area. The formula is as follows:

$$thd(t) = \frac{NDPI(t) - NDPI_{min}}{NDPI_{max} - NDPI_{min}}, \tag{6}$$

where $NDPI(t)$ represents the *NDPI* value at the calendar year date sequence $t$; $NDPI_{min}$ represents the maximum value of *NDPI* in a year; $NDPI_{min}$ represents the minimum value of the time series vegetation curve on the left and right part of the curve (SOS corresponds to the left half of the curve and EOS corresponds to the right) in a year bounded by $NDPI_{max}$; and $thd(t)$ represents the percentage corresponding to $NDPI(t)$ after stretching it in time to a range of 0–1. Critical thresholds of 30% and 70% were selected [34].

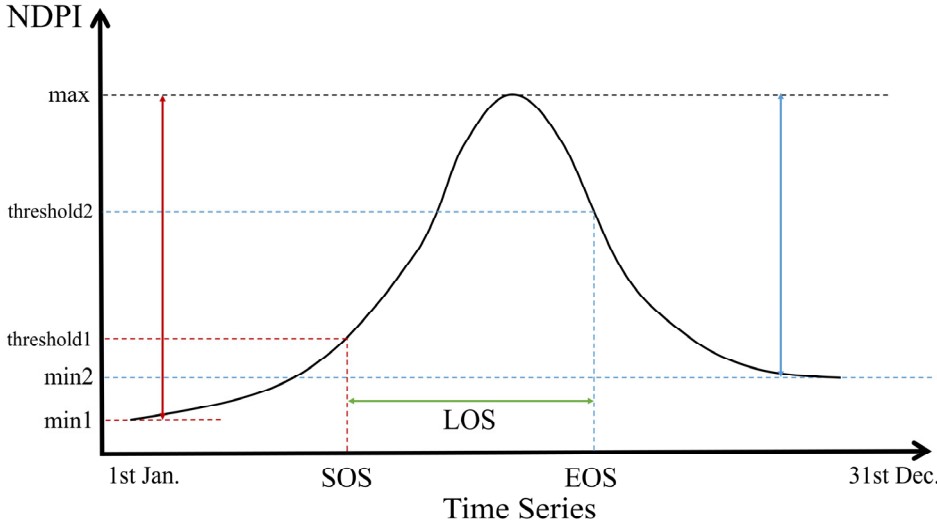

**Figure 4.** Schematic illustration for extracting LSP metrics using dynamics threshold method. The black curve represents the interannual NDPI for a given pixel. Red lines represent SOS, blue lines represent EOS and green lines represent LOS.

*2.5. Correlation Analysis*

To assess the relationship between snowpack and land surface phenology, we calculated the Pearson correlation coefficients between four snow season indicators (FSD, LSD, SSL and SCD) and three phenology indicators (SOS, EOS and LOS), and evaluated the significance of the correlation (*p*-value) by performing a *t*-test [36,65]. Based on the vegetation distribution, we focused on the correlation between snow season metrics and phenological metrics in six different vegetation types: forest, shrub, grass, alpine vegetation, cultivated vegetation and desert vegetation. In addition, an elevation gradient was set from 2500 m to 4500 m to study the vertical variation of the correlation between vegetation and snow seasonality. Correlations were calculated for each pixel and then further counted according to different elevations or vegetation types. The spatial patterns of snow seasonality and land surface phenology over the study area were characterized by the mean of the metrics.

## 3. Results

### 3.1. Spatial Pattern of Snow Seasonality over the QLMA

The spatial heterogeneity pattern of snow seasonality metrics can be observed in the QLMA (Figure 5). The earliest snowfall occurs in the central area within the QLMA (before October, DOY < 30). This region is also the one with the last snowmelt (after April, DOY > 210), resulting in the longest snow season. For the western and northern regions, the first snowfall occurs weeks later (before December, DOY < 90), and snowmelt occurs late (after April, DOY > 210), the length of the snow season is shorter than in the center. In the northern part of Qinghai Lake and the eastern and southern fringes of the study area, the first snowfall occurs the latest (after December, DOY > 120), the snowmelt occurs earliest (before February, DOY < 150), and the snow season is the shortest. The spatial pattern of SCD is similar to that of SSL, but the spatial differences are not significant because of the widespread presence of intermittent snow. The SCD is less than 30 d in most of the study area, especially in the southeast, but higher in the central and western parts.

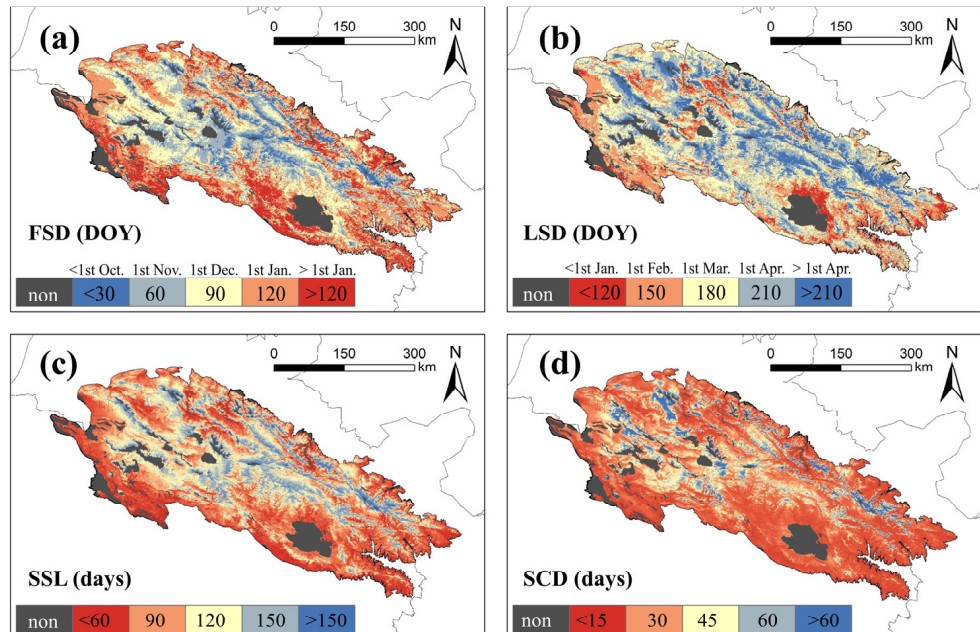

**Figure 5.** Mean of snow seasonality metrics in QLMA from 2002 to 2020. (**a**) represents the FSD, (**b**) represents the LSD, (**c**) represents the SSL, and (**d**) represents the SCD.

### 3.2. Land Surface Phenology among Different Vegetation Types

Figure 6 shows the calculated LSP metrics for the different vegetation types. LSP metrics for all vegetation except desert varied with elevation: SOS delays (Figure 6a), EOS advances (Figure 6b) and LOS shortens (Figure 6c) as elevation rises. There are no significant trends in SOS or LOS with elevation for desert, but EOS advances with elevation.

Figure 6d shows the average LSP metrics across elevation. Desert and alpine vegetation have the earliest growing seasons starting in late April. The growing season of forest begins in mid-May, while shrub, grass and cultivated vegetation start growing in late May. The EOS of different vegetation types are relatively close. The EOS of vegetation types other than alpine vegetation generally appear in early September, with desert and forest a few days earlier. The EOS of alpine vegetation is the earliest, occurring around mid-August. Desert has the longest growing season lengths (SSL = 120), SSL in forest (SSL = 105) and alpine vegetation (SSL = 101) are shorter. Shrub, grass and cultivated vegetation have the shortest growing seasons of about 95 d.

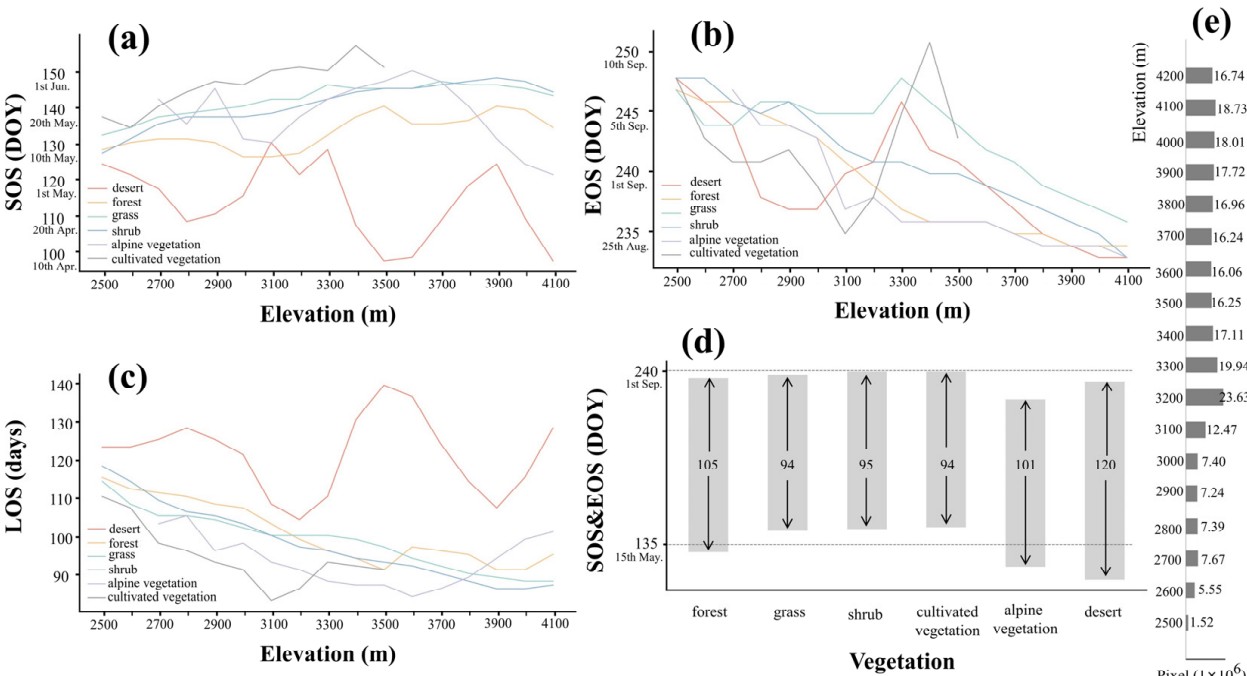

**Figure 6.** The calculated LSP metrics for the different vegetation types change with elevation. (**a**) represents the SOS, (**b**) represents the EOS and (**c**) represents the LOS. (**d**) shows the average of all elevations. The bar graph represents the average SOS and EOS of six different vegetation types, and the arrows inside the bar indicate their LOS. (**e**) shows the histogram of pixel numbers of different elevation gradients.

### 3.3. Spatial Pattern of Land Surface Phenology over the QLMA

The spatial pattern of the mean values of land surface phenology metrics in the QLMA area is shown in Figure 7. Many western areas were filtered out because of the lack of seasonal vegetation. The earliest start of the growing season is in the western and northern margins, usually before 1st May, followed by the eastern areas, where the growing season starts within about a month (before 1st June). Vegetation in the central part of the study area has the latest start of the growing season, occurring after 1st June, and in a few areas even later (after 15th June) (Figure 7a). The growing season for most of the vegetation in the study area ends between 15th August and 15th September. Growing seasons in the northern part of Qinghai Lake and the eastern edge of the study area end half a month later. The vegetation with the latest end of the growing season is located in the western region, occurring after 1st October (Figure 7b). The spatial distribution pattern of LOS is similar to that of SOS, with the northern edge and the sporadic areas in west having the longest vegetation growing season, which exceeds 150 d at most. From the eastern area toward the center, the vegetation growing season gradually shortens from up to 120 d to up to 90 d (Figure 7c).

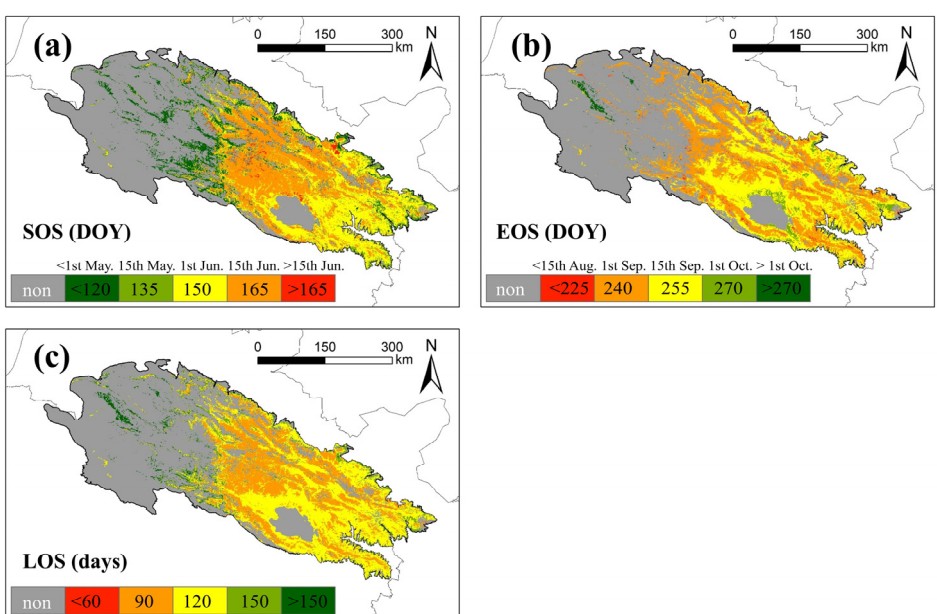

**Figure 7.** Mean of land surface phenology metrics in QLMA from 2003 to 2021. (**a**) represents the SOS, (**b**) represents the EOS and (**c**) represents the LOS.

### 3.4. Spatial Pattern of the Correlation between Snow Seasonality and Land Surface Phenology Metrics

Different snow seasonality metrics have different strengths and directions of influence on different LSP metrics, and there is spatial variation in this correlation (Figure 8). Overall, the snow seasonality metrics have a similar impact on EOS and LOS, in contrast to SOS. FSD shows a significant positive correlation with SOS in the southern part of Qinghai Lake and a mostly nonsignificant negative correlation with the central part. FSD shows mainly negative correlation with EOS and LOS in the study area, especially in the central region, where the negative correlation is significant. EOS and LOS in the eastern part of the study area show a nonsignificant positive correlation with FSD. The effect of LSD on LSP is different from that of FSD. LSD shows a significant negative correlation with SOS in the western part of the study area and the southern part of Qinghai Lake. Both EOS and LOS in the central and western parts of the study area show positive correlations with LSD, where LOS in the western part shows a significant positive correlation with LSD. EOS and LOS in the southeast of the study area, however, show a negative but insignificant correlation with LSD. The spatial pattern of SSL and SCD effects on LSP is similar to that of LSD, compared to the more significant correlation of SCD with LSP.

The proportion of significant positive and significant negative correlations indicates the main direction of influence of snow seasonality metrics on LSP metrics (Table 1). All the 12 correlations have a relatively obvious directionality, meaning that there is no positive correlation with the same proportion of negative correlations. Among them, positive correlations dominate in FSD_SOS, LSE_EOS, LSD_LOS, SSL_L0S and SCD_LOS. The proportion of insignificant positive correlations is about 7% more than the proportion of insignificant negative correlations, and the proportion of significant positive correlations exceeds the proportion of significant negative correlations by twice. However, negative correlation is the major of FSD_LOS, SSL_SOS and SCD_SOS. The proportion of insignificant positive correlations is approximately 6% less than the proportion of insignificant negative correlations, and the proportion of significant positive correlations is less than half of the proportion of significant negative correlations.

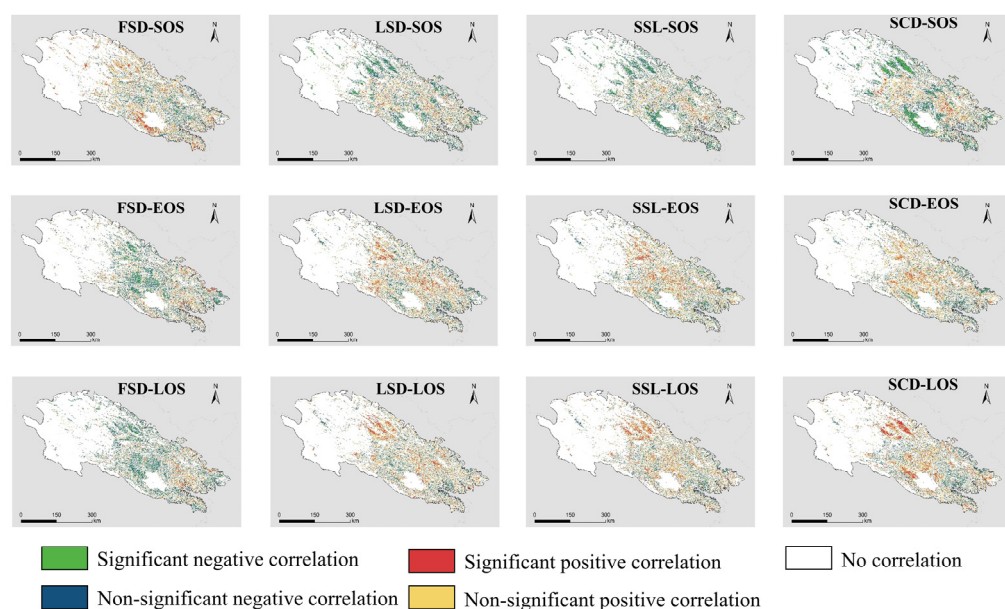

Significant negative correlation　　Significant positive correlation　　No correlation

Non-significant negative correlation　　Non-significant positive correlation

**Figure 8.** Correlation between different land surface phenology metrics and different snow seasonality metrics. "Significant" means $p < 0.1$.

**Table 1.** Area of significant correlation between snow season metrics and phenology metrics (%), SN for significant negative ($p < 0.1$) correlation and NN for nonsignificant negative ($p > 0.1$, $r < -0.2$) correlation. SP for significant positive ($p < 0.1$) correlation and NP for nonsignificant positive ($p > 0.1$, $r > 0.2$) correlation.

|  | SN (%) | NN (%) | NP (%) | SP (%) | SN/SP |
|---|---|---|---|---|---|
| FSD_SOS | 4.23 | 11.75 | 17.79 | 8.96 | 0.47 |
| FSD_EOS | 8.12 | 16.68 | 12.98 | 5.17 | 1.57 |
| FSD_LOS | 9.04 | 18.66 | 11.38 | 4.34 | 2.08 |
| LSD_SOS | 8.89 | 16.66 | 12.54 | 4.91 | 1.81 |
| LSD_EOS | 4.13 | 11.40 | 18.38 | 8.86 | 0.47 |
| LSD_LOS | 4.04 | 11.14 | 18.52 | 9.55 | 0.42 |
| SSL_SOS | 9.66 | 17.22 | 11.47 | 4.34 | 2.23 |
| SSL_EOS | 4.84 | 11.35 | 19.02 | 8.31 | 0.58 |
| SSL_LOS | 3.87 | 10.94 | 18.71 | 10.00 | 0.39 |
| SCD_SOS | 11.88 | 17.31 | 12.90 | 5.26 | 2.26 |
| SCD_EOS | 4.57 | 11.30 | 18.87 | 7.63 | 0.60 |
| SCD_LOS | 3.94 | 11.49 | 18.29 | 11.13 | 0.35 |

*3.5. Elevation-Dependent Correlation between Snow Seasonality and Land Surface Phenology Metrics*

We further investigated the phenological response along the elevation gradient (Figure 9). FSD is significantly negatively correlated with SOS and positively correlated with LOS at low elevation, and the correlation gradually decreases as the elevation increases to 3500 m. The direction of the effect of FSD above 3500 m changes. The correlation with SOS turns from a nonsignificant positive correlation to a significant positive correlation by degrees, and the correlation with LOS changes to a progressively increasing negative correlation. The correlation between FSD and EOS is weak and always fluctuates around 0. There is no shift in the correlation between LSD and LSP metrics with rising elevation. Regardless of the elevation, LSD is always nearly significantly positively correlated with EOS and LOS. LSD is significantly negatively correlated with SOS at low elevation, and the correlation and significance decrease with rising elevation, but there is an increasing trend above 3900 m. The correlation of SSL and SCD with LSP metrics has similar characteristics

with elevation. Below 3300 m, they show a nonsignificant weak correlation with LSP metrics. The correlations gradually increase with elevation, and approach stability at around 3500 m. Above 3500 m, SSL and SCD show a significant negative correlation with SOS and a significant positive correlation with EOS and LOS, especially LOS. The effect of SCD is more obvious than SSL.

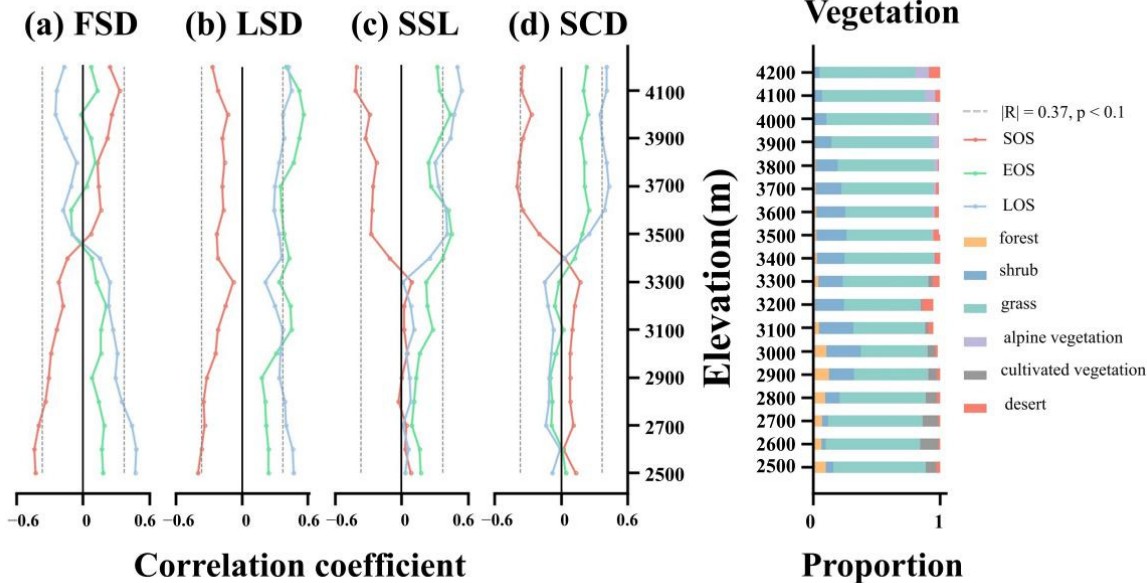

**Figure 9.** Left panel shows the land surface phenology metrics response to different snow seasonality metrics at different elevation. (**a**) means FSD, (**b**) means LSD, (**c**) means SSL and (**d**) means SCD. Right panel shows the vegetation constitute at different elevations.

Vegetation constitute varies at different elevations. Forest and cultivated vegetation are largely absent above 3000 m. Shrub increases rapidly from 4.95% to a maximum of 27.56% above 2800 m, with most of its distribution concentrated between 2900 m and 3700 m. Grass is the dominant vegetation type in most areas, usually accounting for more than 60% of the vegetation.

### 3.6. Interspecific Variation in the Response of Land Surface Phenology

To characterize differences in the responses of land surface phenology to snow seasonality, we compared the responses of six vegetation phenological characteristics to snow (Figure 10). The elevation interval with the highest concentration of vegetation distribution is selected: 2600–3500 m for forest and cultivated vegetation, 3100–4000 m for grass, shrub and desert, and 3600–4200 m for alpine vegetation. FSD is significantly negatively correlated with SOS of forest and LOS of alpine vegetation, and significantly negatively correlated with EOS of shrub. EOS of forest, shrub and grass is significantly positively correlated with LSD, as is LOS of shrub and grass. SSL and SCD affect vegetation in almost the same direction. SSL is significantly correlated with EOS for more vegetation types, while SCD is significantly correlated with SOS and LOS of vegetation. The SOS of forest is significantly positively correlated with SCD, while the SOS of desert is significantly negatively correlated with SCD. EOS of shrub and grassland is more sensitive to SSL and is significantly positively correlated. LOS of alpine vegetation is significantly positively correlated with both SSL and SCD, while LOS of desert is significantly positively correlated with SCD only.

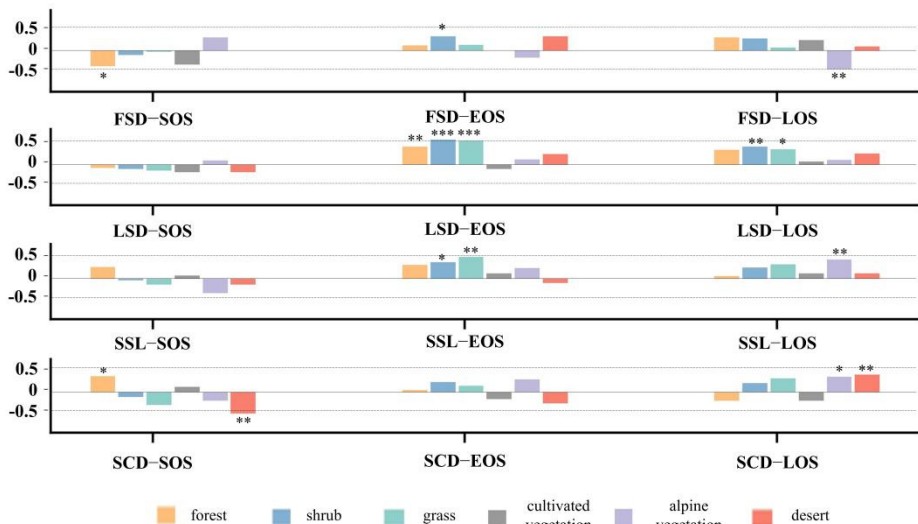

**Figure 10.** Correlation between vegetation phenological metrics of eight types of vegetation and snow phenological metrics. * $p < 0.1$, ** $p < 0.05$, *** $p < 0.01$.

## 4. Discussion

### 4.1. Does Seasonal Snow Seasonality Metrics Affect Land Surface Phenology Metrics?

Our findings demonstrated that land surface phenology at high elevations responds to snow cover seasonality. However, this response relationship is complicated because different snow season metrics affect different phenological metrics (Figure 8, Table 1). The magnitude asymmetry in the significant correlation between the snow seasonality and land surface phenology metrics suggests that a more snow-prone non-growing season (earlier first snow, later snowmelt, longer snow season and more snow cover days) may benefit a more flourishing vegetation growing season the following year (earlier start and later end of the growing season, longer growing season). The area of significant positive correlation between FSD and SOS is more than two times the area of significant negative correlation. In contrast, the area significantly positively correlated with LOS is less than half of the area significantly negatively correlated. These magnitude asymmetries may indicate the direction of the effect of FSD on LSP [33]. That is, FSD is positively correlated with SOS and negatively correlated with EOS and LOS. This suggests that earlier snowfall in the autumn triggers an earlier growing season the following spring and delays the end of the vegetation growing season the following autumn, which also extends the vegetation growing season.

Qiao and Wang [38] found no or negative correlation between FSD and SOS when exploring winter snowpack and spring grassland vegetation phenology in Inner Mongolia, which is not consistent with our findings. This may be attributed to elevation. Significant negative correlations could be observed only in central Inner Mongolia, where the elevation is below 3000 m. However, at higher elevations in the southwest, the correlations are not significant. In QLMA, there are relatively few areas below 3000 m in elevation, resulting in the negative correlation that is not widely observed by us. The findings of Wang et al. [35] in the Tibetan Plateau (TP) are close to ours. They extensively observed a positive correlation between FSD and SOS, especially in the central TP and northwest of the QLMA. The negative correlation between FSD and EOS is consistent with the findings of Qi et al. [34] in QLMA; moreover, we observed a larger area (Figure 8). This finding is equally relevant in other areas with continuous seasonal snowpack [66]. However, the impact of late season transient snowfall events may be limited, which could explain the small impact of FSD in some areas [58,67].

We considered that a later snowmelt may result in a delayed end of the growing season and a longer growing season length. Longer snow seasons and more snow days lead to earlier growing seasons and also prolong growing season length. Our study confirmed the findings of Wang [35] and Qi [34] in the same study area, but differences in elevation and

hydrothermal conditions cause this finding to change in other areas. In fact, the effect of different snow seasonality metrics can still be observed even within a region, which may also be due to the elevation and climatic conditions of the different regions [36]. Moreover, different vegetation indices, LSP estimation methods and thresholds for estimating snow seasonality have an impact on the conclusions. However, these effects are usually reflected in intensity and significance. The apparent magnitude asymmetry of correlations within the study area points to the main direction of snow seasonality effects, in which we are able to corroborate each other. Even in different areas, similar effects of snow seasonality can still be observed if elevations and climates are similar, such as at higher elevations in Inner Mongolia and Nepal [38,66].

*4.2. Why Do the Effects of Snow Seasonality Metrics Vary with Elevation?*

Our study confirmed the high-elevation dependence between snow cover and phenology (Figure 9). The effect of earlier FSD on SOS shifted from delaying to facilitating with increasing elevation. The effect on LOS changes from shortening to lengthening, with the turning point occurring roughly at 3500 m. The correlation between FSD and EOS changes from a nonsignificant positive correlation to almost no correlation. If an earlier and longer growing season is considered to be better, then an earlier FSD is detrimental to vegetation growth at lower elevations and beneficial at higher elevations. This effect of first snow on vegetation phenology with elevation is not unique to QLMA. Paudel and Andersen [66] found no correlation between FSD and EOS at low elevations, but a strong positive correlation at very high elevations. Obviously, the elevation of our study area is far from 'very high', above 5000 m, so we could only observe an insignificant correlation between FSD and EOS. The findings in the Qinghai–Tibetan Plateau and QLMA are more comparable. Qi [34] divided QLMA into four elevation intervals, and the correlation between FSD and EOS is consistent with our findings on the trend of weakening with elevation. Our more detailed division makes the results more obvious. Wang et al. [35] found a positive correlation between FSD and SOS on the TP that gradually increased between 2500–5000 m and then weakened, which is similar to our findings. We further found a change in this correlation not only in intensity but also in direction, that is, a significant negative correlation between FSD and SOS below 3500 m. The study area of Wang et al. [35] is much larger than ours, and the seasonality of snow varies greatly at lower elevations, which may lead to their inability to accurately count correlations below 3500 m. However, that earlier first snowfall at higher elevations favors earlier vegetation phenology is what we all agree on.

We did not observe an effect of LSD on LSP metrics with elevation. Regardless of elevation, later snowmelt is beneficial for earlier vegetation growing season initiation and a longer growing season. Snow melt directly provides the necessary water for vegetation to sprout, and the spring snowpack maintains soil temperatures. No matter what the elevation, accumulated temperature and water are necessary for vegetation to sprout. Paudel and Andersen [66] observed the same conclusion as ours in the low elevation arid zone, and Wang et al. [35] also found a negative correlation between LSD and SOS in the TP, but it was not significant at low elevation. In contrast, Qi et al. [34] found a significant positive correlation between LSD and SOS in the high-elevation interval of QLMA, which may be related to the selection and treatment of the vegetation indices. Compared to estimating SOS based on NDVI, the NDPI we used is shown to better eliminate the effect of pre-season snowpack and avoid misclassification of snow and vegetation pixels [40–42]. In addition, our reconstructed NDPI with a temporal resolution of 4 d also helps to obtain a more accurate SOS.

We also found a similar effect of SSL and SCD on LSP metrics. Vegetation below 3300 m is barely affected by them, while vegetation above 3500 m SOS is significantly negatively correlated with them, and LOS is significantly positively correlated with them. This is consistent with the findings of many other studies, at least at the same elevation [34,35,58]. This may be due to different climatic conditions at different elevations, which could drive differences in correlation [68]. Although the melting of snow will always provide moisture

to the soil, different temperature conditions may lead to different effects on the presence of snow. More snow is needed at higher elevations to cover the soil than at relatively warm lower elevations because the snow acts as an insulator [36,69]. Soils are protected from the harsh climatic conditions and severe solar radiation to which they would otherwise be subjected by a snow cover [69–71]. Soil temperatures at high elevations with snow cover are usually higher than in areas without snow [72].

We described for the first time at QLMA the process of reversal of the direction of influence of snow seasonality metrics on LSP metrics with elevation. However, we estimated that the thresholds for LSP (0.3 and 0.7) and the use of seasonal vegetation filter (NDPI > 0.1) may have influenced the strength of the relationship. The vegetation constitute that varies with elevation may be another factor affecting the correlation. Changes in other vegetation proportions can cause fluctuations in correlations (LSD_FSD below 3300 m), but grass is always the predominant vegetation type in QLMA, which could ensure the relative stability of correlations.

### 4.3. Why Do the Effects of Snow Seasonality Metrics Vary with Vegetation Type?

The response of vegetation to snowpack varies considerably between biomes, and similar phenomena have been observed in the QLMA (Figure 10). We found that the LSP metrics of shrub and grass respond most significantly to LSD compared to other vegetation, namely that later snowmelt extends the growing season of both vegetation species. This is consistent with the findings of many other studies on grasslands [34,36]. It may be because shrub and grass have relatively simple structures and short size, so they are more likely to be completely covered by snow. In addition, due to the strong solar radiation, the snow on the grassland melts more easily and water can be supplied to the soil in a timely manner [73]. The LSP metrics of desert only respond to SCD. Compared to other snow seasonality metrics, SCD reflects not only the timing of snow presence, but also the frequency of snow presence, which is important for drought desert. Conversely, alpine vegetation is more sensitive to FSD and SSL and these may reflect the arrival and duration of cold air during the snow season as temperature is more important for alpine vegetation. We also found that the SOS of the forest responds differently to snow seasonality metrics than other vegetation types. Snow falling on branches does not have a direct and timely effect on the root system [74,75]. Due to the tall structure of the trees, the snow in the canopy and on the ground is exposed to different solar radiation, resulting in differences in snow melt [76,77]. Furthermore, some studies have demonstrated that different vegetation types have different temperature requirements for breaking dormancy [69,78]. Woody plants may require cold conditions to promote germination, while grasses require warmer conditions. Snow protects shallow underground root systems from the cold, resulting in different effects on different vegetation types.

In summary, our study illustrated that different vegetation types have different reflections of LSP metrics on snow seasonality metrics, which explains the spatial variation in the impact of snow seasonality metrics from another perspective than elevation.

### 4.4. Prediction of Vegetation Phenology from Satellite Data Is Beneficial for Future Research

The concept of using satellite data to estimate vegetation phenology metrics was born long ago. With the continuous improvement of remote sensing image accuracy and cloud computing capability, the estimation of phenology based on satellite data has gradually become reliable. Unlike the phenology metrics obtained from field observations, the remote sensing-based estimation of phenology metrics focuses on the variation of regional greenness. Important time points are calculated from the interannual variation curves of vegetation indices, and thus vegetation phenology is estimated. For stakeholders, satellites can quickly provide long time series and large-scale data, which could also avoid time-consuming field work. Characterizing vegetation phenology at a larger scale is beneficial for studies such as vegetation response in the context of global climate change.

However, the validation for the remote sensing-based phenological indices and in situ phenological indices is also important. Although there may be temporal differences in the phenological indices obtained by the two methods, both can indicate consistent phenological trends. More in situ data is an important way to improve the accuracy of satellite predictions. This is also the objective of our future study.

*4.5. Study Limitations and Future Work*

The medium-resolution satellite data and complex topographic conditions of the Qilian Mountains make it difficult to estimate land surface phenology. Although we used the NDPI, which is least affected by pre-season snowpack, as a vegetation index, the accuracy of such threshold-based extraction remains uncertain. Land surface phenology is a complex parameter that is influenced by a combination of factors. In addition to snow seasonality, vegetation is influenced by snow depth, pre-growing season temperature and precipitation and light conditions. In addition to elevation, slope and aspect play important roles. Thus, assessing the response of land surface phenology requires more accurate models. As mentioned earlier, the differences in water conditions across the study area may be an important factor influencing the conclusions. The content and depth of groundwater, meltwater from permafrost and glaciers, may also have an impact. These would be good to include in future study. Finally, although the vegetation distribution data we used are very reliable, changes are inevitable over long time series, especially in low-elevation areas that are inherently more susceptible to human activities.

## 5. Conclusions

The snow season and vegetation phenological indicators in the Qilian Mountains in the northeastern Qinghai–Tibet Plateau were investigated, and their corresponding relationships were analyzed. In this study, we concluded that snow seasonality metrics have distinct spatial distribution characteristics. The snow season started earlier and lasted longer in the central part of the study area. The LSP metrics varied significantly with elevation and most vegetation growing seasons shortened with elevation. The asymmetry of significant correlation between snow seasonality and LSP metrics indicates the main direction of influence. A more snow-prone non-growing season (earlier first snow, later snowmelt, longer snow season and more snow cover days) may trigger a more flourishing vegetation growing season the following year (earlier start and later end of growing season, longer growing season).

The NDPI we used is less affected by spring snowpack. We set thresholds to remove nonseasonal vegetation and delineated more detailed elevation gradients. We described the effect of QLMA snow seasonality as a curve that varies with elevation. Below 3300 m, later first snowfall leads to an earlier growing season and also extends the growing season length, while the effect of first snowfall above 3300 m is reversed. The intensity of the effect of LSD fluctuates with elevation but does not reverse. The effects of SSL and SCD on LSP are small and insignificant below 3500 m, and their increase mainly benefits the extended growing season of high-elevation vegetation. In addition, the sensitivity of LSP metrics to snow seasonality varies among vegetation types. Our research provides more evidence that the impact of snow varies with elevation and underlying vegetation types.

Hydrothermal conditions, changes in temperature and precipitation, extreme weather events and glacial melt are important factors influencing land surface phenology at high elevations and should be investigated in future studies in conjunction with high-resolution data to develop improved models for analyzing them.

**Author Contributions:** Conceptualization, W.Z., K.Y. and Y.L.; methodology, Y.L. and K.Y.; software, validation, Y.L. and K.Y.; writing—original draft preparation and visualization, Y.L.; writing—review and editing, X.M. and S.G. funding acquisition, W.Z. and K.Y. All authors have read and agreed to the published version of the manuscript.

**Funding:** This research was funded by the National Natural Science Foundation of China, grant number 41977415, and the Fundamental Research Funds for the Central Universities, grant number 265QZ2022001.

**Data Availability Statement:** Not applicable.

**Acknowledgments:** We express our gratitude to anonymous reviewers and editors for their professional comments and suggestions.

**Conflicts of Interest:** The authors declare no conflict of interest.

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
