# Peer review of "Phenological Responses to Snow Seasonality in the Qilian Mountains Is a Function of Both Elevation and Vegetation Types"

_remotesensing, doi:10.3390/rs14153629_

Round 1

Reviewer 1 Report

The authors have sufficiently addressed most of my previous comments. One thing I would like to point out is that regarding my comment 3 “I would like to see the authors further discuss the implication of accurate plant phenology prediction based on satellite images”, I do not think the authors have answered my question. Assuming and knowing plant phenology can be accurately predicted by satellite images, how can this methodology be applied and how would it be beneficial to stakeholders? Also Figure 3 is wrong.

Author Response

Point 1: The authors have sufficiently addressed most of my previous comments. One thing I would like to point out is that regarding my comment 3 “I would like to see the authors further discuss the implication of accurate plant phenology prediction based on satellite images”, I do not think the authors have answered my question. Assuming and knowing plant phenology can be accurately predicted by satellite images, how can this methodology be applied and how would it be beneficial to stakeholders? Also Figure 3 is wrong.

Response 1: Thank you for your constructive suggestions, and we apologize for not answering this question accurately in our previous manuscript.

In the Revised manuscript, we added 4.4 Prediction of vegetation phenology from satellite data is beneficial for future research in 4. Discussion. We would like to show the reader that satellite-based estimates of vegetation phenology metrics are convenient and feasible. For stakeholders, the use of satellite estimates is a good option if in situ data are not available for observation. The metrics predicted by satellites could characterize vegetation phenology on a larger scale, which may be also useful for larger scale studies.

We also rechecked Figure 3 and found the error in the formula and capitalization. We apologize for our mistake!

Thank you again for your comments and we appreciate your professionalism and patience.

These changes can be found on page 16 in the 4.4. Prediction of vegetation phenology from satellite data is beneficial for future research in LN 516 -533 and page 5 in Figure 3. Overview of technical workflows.

Reviewer 2 Report

I suggest the manuscript can be accepted after English improvement

Author Response

Point 1: I suggest the manuscript can be accepted after English improvement. 

Response 1: Thank you for your comments on our manuscript. In the Revised manuscript we rechecked the grammar and spelling and tried to correct some inappropriate expressions.

Thank you again for all the help you have ever provided and we appreciate your professionalism and patience.

Reviewer 3 Report

In this study, to refine the knowledge on how snow phenology affects the vegetation phenology in a mountainous region, the authors explored how vegetation phenology and this influence varies among different vegetation type and elevations. They extracted the vegetation phenology including SOS and EOS based on NDPI, which is a well performed vegetation indices in snow covered regions. The Qilian Mountains can be a specific region to be studied with various types of vegetation. The results showed the change of vegetation phenology is varied along with altitude and vegetation type. The study is interesting and is well presented in general, but still needs some improvement before publishing. The writing also need be polished before publishing.

 Point 1: Thanks for the nice illustration of snow cover’s influence on vegetation phenology. It makes the background of why snow cover can be vital to phenology clear to the readers. However, there are still some point are missing. First, the introduction did not state the significance of the Qilian Mountains in the phenology study. As stated by the author, Qilian Mountain is a very small region of the Tibetan Plateau, in which there have been a lot of similar studies conducted. Why Qilian Mountain itself can be very special in the phenology study? Second, the ‘vegetation index’ and ‘vegetation indices’ are different, please note the difference.

Point 2: The workflow in ‘materials and methods’ also benefit the understanding of this research. We appreciate the authors using 500m remote sensing products to estimate the phenology change in the high mountain region.

A suggestion is that the validation for the remote sensing based phenology indices and the in situ phenology indices might be needed here.

Why only use the correlation analysis instead of the partial correlation analysis for multiple variable relationship analysis?

Point 3: Result section, though most of the figures are relevant and well presented, some part is still missing.

The author shows the spatial pattern of different snow phenology indices and the phenology indices, but the author did not present the trend of snow and vegetation phenology. Is this also a vital part for the reader to understanding why the research is necessary in Qilian Mountain? Are the trends of vegetation phenology over the Qilian Mountains varies among different altitude gradients?

Why using 3300m as the thresholds for altitude gradients, try add the histogram of pixel numbers of different altitude gradients in Figure 6.

Point 4. Discussion section, I appreciate the author compared this study with some recent relevant researches, and make several explanations for the inconsistence of this study and other studies. More fundamental reasons for the difference of snow’s impact might be the water condition in alpine and Inner Mongolia plateau, this can be noted in the future studies. This can benefit the readers’ future work.

Some detailed suggestions for the grammar mistakes and typos:

1.     Abstract line 23: 2) Vegetation phenology at high elevations is sensitive to the length of the snow season and the number of snow cover days, hardly affecting below 3500 m.

2.     Line 369  ‘the magnitude asymmetry’

3.     Line 454 ‘more snow is needed’

4.     Line 21, line 370, 372, 514 ‘positive snow season’ ‘positive growing season’ seems confusing and improper, maybe replaced by ‘more snow-prone non-growing season (earlier first snow)’. ‘Tigger’ sounds too strong, might be ‘benefit’

5.     Note the difference of ‘vegetation indices’ and ‘vegetation index’ (Normalized Difference Vegetation Index).

Author Response

Point 1: Thanks for the nice illustration of snow cover’s influence on vegetation phenology. It makes the background of why snow cover can be vital to phenology clear to the readers. However, there are still some point are missing. First, the introduction did not state the significance of the Qilian Mountains in the phenology study. As stated by the author, Qilian Mountain is a very small region of the Tibetan Plateau, in which there have been a lot of similar studies conducted. Why Qilian Mountain itself can be very special in the phenology study? Second, the ‘vegetation index’ and ‘vegetation indices’ are different, please note the difference.

Response 1: Thank you for your constructive suggestions, we will answer your questions from three aspects.

Firstly, it has been shown that the Tibet Plateau (TP) is a very sensitive region to climate change, as is the Qilian Mountaina area in the northeastern part of the TP. Researches on the TP, particularly in the Qilian Mountains, focused more on ecosystem structure and function, but less on the response and changes in vegetation phenology. However, changes in vegetation phenology could also lead to changes in ecosystems, which might easily be overlooked. The Qilian Mountain is an important ecological barrier in western China, and phenological studies are good complements to others.

Secondly, as you commented, there have been many studies of the TP, but the large size of the area makes conclusions spatially variable and difficult to generalise. The smaller size of the Qilian Mountains area may reduce the spatial interference with the conclusions.

Finally, of course, not all small areas on the Tibetan Plateau can be considered as ideal study areas. The Qilian Mountains area is the source of several rivers and contain a large number of glaciers and, more importantly, a rich variety of vegetation types. In addition, the Qilian Mountains area have a clear elevational gradient from 1719 m at the edge to 5814 m in the centre. These provide the conditions for discussing the response of vegetation phenology under different conditions. Therefore, we chose the Qilian Mountains as a representative area to study the vegetation phenology response to snow seasonal.

We apologise for neglecting to state the above in the introduction and your comments are very crucial. In the Revised manuscript, we added a statement about the study area in the introduction. We also apologise for the incorrect words, and we corrected the expression in the Revised manuscript. Thank you again for your constructive comments.

These changes can be found on page 3 in the 1. Introduction in LN 98 -104.

Point 2: The workflow in ‘materials and methods’ also benefit the understanding of this research. We appreciate the authors using 500m remote sensing products to estimate the phenology change in the high mountain region.

A suggestion is that the validation for the remote sensing based phenology indices and the in situ phenology indices might be needed here.

Why only use the correlation analysis instead of the partial correlation analysis for multiple variable relationship analysis?

Response 2: Thank you for your comments and questions, we would like to answer them separately.

As you have commented, it is essential to use in situ data to verify the accuracy of remotely sensed data. Vegetation phenology based on remote sensing estimates is defined as land surface phenology. It shows the change curve of greenness in the region. Significant points in the curve can be selected by function fitting or thresholding to represent phenological metrics of the vegetation. In contrast to observations by professionals or phenological cameras, estimates based on remote sensing could not give a clear date of the sprouting or leaf spreading of a particular plant, but the trends are consistent. Our satellite based estimates of phenology are cognitively consistent in time (specific phenological periods) and space (variation of phenological periods with elevation). Fewer phenological stations and the difficulty of obtaining data are the main obstacles. Also due to time constraints we are sorry that we did not have field observations. In a further study we will try to observe in the field and validate using multiple sources of data.

Your comment on the methods of analysis is very important. In previous studies, the influences of snow seasonality metrics on vegetation phenology have been studied using different correlation analysis methods. The Pearson correlation coefficient is a simple and commonly used correlation coefficient that has been widely used in previous studies. Many studies demonstrated that Pearson correlation coefficients coould effectively characterise the effect of snow seasonality metrics on vegetation phenology. Considering the limitations of Pearson correlations, we further compared the proportion of positive and negative correlations. This magnitude asymmetry is a more important basis for our judgement, which might compensate for methodological shortcomings. In future studies we will further research the impact of different correlation analyses on the results.

Thank you again for your specific and professional questions.

These changes can be found on page 16 in the 4.4. Prediction of vegetation phenology from satellite data is beneficial for future research in LN 516 -533.

Point 3: Result section, though most of the figures are relevant and well presented, some part is still missing.

The author shows the spatial pattern of different snow phenology indices and the phenology indices, but the author did not present the trend of snow and vegetation phenology. Is this also a vital part for the reader to understanding why the research is necessary in Qilian Mountain? Are the trends of vegetation phenology over the Qilian Mountains varies among different altitude gradients?

Why using 3300m as the thresholds for altitude gradients, try add the histogram of pixel numbers of different altitude gradients in Figure 6.

Response 3: Thank you for your constructive comments. We would like to answer your question separately.

We showed the trends in snow seasonality metrics and vegetation phenology in a more original manuscript. We calculated the standard deviation of each pixel over the last twenty years and also examined whether there was a clear trend by fitting curves. However we find that these may be too much. However, we found that these may be too much. We focused on the response of vegetation phenology variation to changes in snow seasonality metrics. The difference in correlation across situations is of more interest to us in this study than the interannual variability of single factors. Therefore, we considered removing this section. “The trends of vegetation phenology over the Qilian Mountains varies among different altitude gradients”. It is a very worthwhile study, which can be combined with the previously mentioned elements, and we will continue to study them in the future.

As Fig. 9 shows, the effects of FSD, SSL and SCD on vegetation phenology metrics were reversed in the 3300 m to 3500 m interval. This may indicate that there is significant elevational heterogeneity in the effect of snow on vegetation phenology in the QLMA. The threshold probably occurs in the interval from 3300 m to 3500 m, with the lowest elevation near 3300 m being the first time such a reversal can be observed (SSL).

Thank you for your comments regarding Figure 6, and adding a statistical chart is indeed more beneficial. We added a vertical bar chart to the right of Figure 6 which depicts the number of pixels at different elevations.

These changes can be found on page 9 in the Figure 6.

Point 4: Discussion section, I appreciate the author compared this study with some recent relevant researches, and make several explanations for the inconsistence of this study and other studies. More fundamental reasons for the difference of snow’s impact might be the water condition in alpine and Inner Mongolia plateau, this can be noted in the future studies. This can benefit the readers’ future work.

Response 4: Thank you for such constructive comments.

As you suggested, water condition is a more fundamental cause of differences in snow impact, especially in mountainous or highland areas. The depth and content of groundwater in different study areas can change the direction and intensity of snow impacts. The glacial permafrost resources contained in the QLMA are also worth considering. We further strengthen the description of these in our discussion, which may benefit the reader for future research.

These changes can be found on page 16 in the 4.5. Study Limitations and future work in LN 534 -550.

Point 5: Some detailed suggestions for the grammar mistakes and typos:

1.Abstract line 23: 2) Vegetation phenology at high elevations is sensitive to the length of the snow season and the number of snow cover days, hardly affecting below 3500 m.

2.Line 369 ‘the magnitude asymmetry’

3.Line 454 ‘more snow is needed’

  1. Line 21, line 370, 372, 514 ‘positive snow season’ ‘positive growing season’ seems confusing and improper, maybe replaced by ‘more snow-prone non-growing season (earlier first snow)’. ‘Tigger’ sounds too strong, might be ‘benefit’
  2. Note the difference of ‘vegetation indices’ and ‘vegetation index’ (Normalized Difference Vegetation Index).

Response 5: Thank you for such detailed comments, which are very helpful to us. We apologise for inappropriate expressions and errors. In the Revised manuscript we rechecked the grammar and spelling and tried to correct some inappropriate expressions.

Thank you again for all your comments and questions, and we especially appreciate your patience and professionalism. It not only helps with this manuscript, but also provokes us to think about future research.

This manuscript is a resubmission of an earlier submission. The following is a list of the peer review reports and author responses from that submission.

Round 1

Reviewer 1 Report

The study examined the relationship between snow days and growing season of Qilian Mountains using satellite images. The manuscript is generally written in detail. Its research questions are clearly defined. Study results were well presented and discussed in depth.

Regarding the areas where the manuscript can be improved, I would like to start with the usage of the term “phenology” in the manuscript. Phenology is generally used to refer to periodic events in biological life cycles, such as distinct plant growth stages. The study only dealt with the beginning, the end and the length of growing season, rather than actual plant phenological stages or how the vegetations changed during the growing season. In my opinion the usage of the word “phenology” is not very appropriate, although technically not wrong. I recommend the authors changing the wording of the manuscript to simply focus on “the timing and length of growing seasons” rather than “phenology”. Additionally, I do not think the term “snow (cover) phenology” make sense as snow is not a biological life form. I recommend the authors paraphrasing the term.

The introduction of the manuscript can be improved in terms of reviewing other relevant studies using similar data analysis methodologies. A comprehensive, detailed literature review is missing. Aside from the studies mentioned in line 68-69, if there are no other relevant studies, the authors can expand the review on these studies by adding more information regarding their methods and results.

I would like to see the authors further discuss the implication of accurate plant phenology prediction based on satellite images, and how it would benefit the research community in introduction or discussion section.

It would be a good idea to add figures in section 2.2 for illustration.

Reviewer 2 Report

In the attachment, I provided 151 comments that cover all issues of this paper.

Below is a list of the main problems:

1)      Improper citation: Authors use papers from a last few year as a reference for knowledge that was established much earlier. Majority is improper, or some wrongly used or in a wrong place. I must admit that is a bit frustrating, while someone tries to show that none work in this topic has been done earlier.

2)      Data and methods are ill-described.

a.       It is unclear which data layer of the SNOW MODIS product authors used, and if the quality filtering has been done.

b.       Authors referred to wrong products and support they decision based on the studies that used a different product version.

c.       It is not clear how the Soil temperature and soil water data were processed.

d.       I think that there is a mistake in SSL calculation (eq.3)

3)      In Results section:

a.       Authors did not provide significance level for they correlation analysis of snow & lsp while using terms “significant correlation”. That is not acceptable!

b.       Saying that something is “vary significantly” without statistic provided is not very scientific.

4)      The Discussion section is just a collection of snow effects on LSP descriptions from different papers (and again problem with reference: those things were studied much earlier than provided citation) that would partially go to the Introduction. Authors did not compare their results with other papers. Nor have shown how their study goes beyond the actual state of knowledge and what is the new input that extends the understanding.

They cited one paper that is very similar to this one, without the effort to compare both results. I find some statements contradicting the results, or conclusion that are rather assumptions because were not studied/proved here.

5)      In the conclusions, authors said that the study provides ‘further” evidence, but I do not see how further at all.

6)      Besides, there are many language problems: confusing sentences, improper language, spelling mistakes. Also, what is very important; it is snow SEASONALITY not phenology.

7)      Graphics need to be corrected: labels, colors etc.

All the above is in the attached pdf.

To sum up, I do not see how this paper can contribute to filling the knowledge gap (was it even stated?). I do not see any novelty, especially while authors cited a paper that presents very similar research for that area: snow effects on land surface phenology but for slightly different time frame and different vegetation index. Also methodology mistakes and imprecision of results does not allow it to be published.

Reviewer 3 Report

I recommend that the abstract need further carefully edit. Please present the concise abstract as “background and aims”, “methodology”, “results”, and “conclusion” with key and significant points of the authors research. Please be specific and precisely, and the authors need to avoid the general information and less meaningful sentences in this section. In particularly, “methodology” is too short, but “results” is too long, and “conclusion” are too short and lack final key points of the research. Keywords, add “high-elevation” or “alpine vegetation”. First, the introduction section should avoid general and additional information. And the terms and definitions really need carefully check. Again, the authors are focusing on three research questions. These three objections should have tight link with previous introduction content that to avoid weak motivation and less scientific meaning. Section 3. Result. I suggest the authors carefully think about the key results which should be put in this section. Some additional figures could be moved to supporting materials. Section 4. Discussion. The authors should further clarify what the key points indicating based on their finding in this study. Which result is expected or in agreement with previous studied in the field. What is in disagreement with previous study, why? Then provide evidence and arguments. And what is new in the field, the so what? Conclusion, please use the short sentences, long sentences are not good for reading.

Reviewer 4 Report

Liu et al analyze the impact of snow seasonality on land surface phenology for the Qilian mountatins using 18 years of MODIS remote sensing data. While the topic is promising in terms of evaluating the links between snow and phenology, the manuscript has serious methodological flaws and is missing key information in the methodological description. See also comments below.

Major comments:

  • Figure 6 is completely not-understandable. Please do not use rainbow-scales, since they are perceptually bad and can lead to wrong conclusions. Better to use perceptually uniform colour scales, such as viridis, or colorbrewer, or many others. Also provide separate correlations for all metrics (3 LSP x 4 snow), and not combined.

  • Statistical analysis in unclear. 1) Did you calculate pixel-by-pixel correlations? 2) How did treat elevation and vegetation groups? Did you calculate one correlation for all pixels, or average the pixel-by-pixel correlations? If only one correlation per group, then it is highly likely that spatial correlation is confounded with the temporal correlation. How did you apply the test for significant correlations at the group scale? They are not independent observations (spatial correlation).

  • Figure 7 and Table 1 are contradicting. The area with significant correlations is ~10%, but then the correlations in Figure 7 are very high at high elevations. Did you only consider significant correlations in Figure 7 or not?

  • There is no discussion on the very low magnitude of correlations as well as extremely low area of significant correlations. If ~90% of correlations are low and/or insignificant, how meaningful are the results?

  • Table 2 unclear. How did you arrive at these results? What did you compare? The percentages suggest that everything was significantly correlated. In addition, the results are likely overconfident, since I assume you increased resolution from ERA5 to MODIS, and just duplicated values?

  • Section 3.1/Figure 3: How can it be that SSL is 5-10 times higher higher than SCD? Is it all intermittent snow? Related: How do you explain the strongly differing and sometimes contradicting results between SSL and SCD?

Minor comments:

  • L51-53: Inaccurate or wrong. Snow cover insulates in winter, but ground temperatures are at or below freezing, so no “accumulation” of heat possible. Only at snow melt, soil temperatures increase.

  • L74: what negative effect?

  • L134: what do you mean by “corrected and stitched”?

  • L142: I don’t think all the references here give results for the concrete cloudremoval that the authors did. Please separate general results from specific ones to your study.

  • Please define NDPI

  • Section 2.3: Unclear. First you say FSD/LSD are DOY, but then you define it as difference to May 1 or Sep 1. Related: If it’s a difference, it can be negative, too, right?

  • Section 2.4: Unclear how you calculated SOS and EOS.

  • Figure 4: Please do not use two-axis figures.

  • Section 3.2: Vegetation type abbreviations not introduced.